# Integrable quadratic structures in peakon models

J. Avan[1], L. Frappat[2*] and E. Ragoucy[2]

**1** Laboratoire de Physique Théorique et Modélisation, CY Cergy Paris Université,
CNRS, F-95302 Cergy-Pontoise, France
**2** Laboratoire d'Annecy-le-Vieux de Physique Théorique LAPTh ,
CNRS, Université Savoie Mont Blanc, F-74940 Annecy

⋆ luc.frappat@lapth.cnrs.fr

## Abstract

We propose realizations of the Poisson structures for the Lax representations of three integrable $n$-body peakon equations, Camassa–Holm, Degasperis–Procesi and Novikov. The Poisson structures derived from the integrability structures of the continuous equations yield quadratic forms for the $r$-matrix representation, with the Toda molecule classical $r$-matrix playing a prominent role. We look for a linear form for the $r$-matrix representation. Aside from the Camassa–Holm case, where the structure is already known, the two other cases do not allow such a presentation, with the noticeable exception of the Novikov model at $n = 2$. Generalized Hamiltonians obtained from the canonical Sklyanin trace formula for quadratic structures are derived in the three cases.

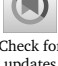

## Contents



# 1 Introduction

Peakon solutions to non-linear two-dimensional $(x, t)$ integrable fluid equations have been shown to exhibit themselves integrable dynamics in several interesting cases. They take the generic form

$$\varphi(x, t) = \sum_{i=1}^{n} p_i(t) e^{-|x - q_i(t)|}, \tag{1}$$

and their dynamics for $(p_i, q_i)$ is deduced from a reduction of the 1+1 fluid equations for $\varphi(x, t)$.

Integrability of peakons entails the existence of a Poisson structure for $(p_i, q_i)$ deduced from the original Poisson structure of the fluid fields, including $\varphi(x, t)$; the existence of a Hamiltonian $h(p_i, q_i)$ and Poisson-commuting higher Hamiltonians, a priori deduced by reduction of the continuous Hamiltonians to peakon solutions for the integrable 1+1 dynamical system. In a number of cases, the dynamics is expressed in terms of a Lax equation:

$$\dot{L} = [L, M], \tag{2}$$

where $L, M$ are $(p, q)$-dependent matrices and (2) contains all equations for $p_i(t)$, $q_i(t)$ obtained from plugging (1) into the 1+1 integrable equation.

The Lax matrix naturally yields candidate conserved Hamiltonians $h^{(k)} = \operatorname{tr}(L^k)$. Poisson-commutation of $h^{(k)}$ is equivalent [1] to the existence of an $r$-matrix formulation of the Lax matrix Poisson brackets:

$$\{L_1, L_2\} = \sum \{L_{ij}, L_{jk}\} e_{ij} \otimes e_{kl} = [r_{12}, L_1] - [r_{21}, L_2]. \tag{3}$$

The $r$-matrix itself may depend on the dynamical variables [2]. A simple example of such dynamical $r$-matrix is given by the reformulation of the well-known "quadratic" $r$-matrix structure, extensively studied [3–5]. We recall the form of this quadratic structure:

$$\{L_1, L_2\} = a_{12} L_1 L_2 - L_1 L_2 d_{12} + L_1 b_{12} L_2 - L_2 c_{12} L_1, \tag{4}$$

where $a_{12} = -a_{21}$, $d_{12} = -d_{21}$, $b_{12} = c_{21}$ to ensure antisymmetry of the Poisson bracket. When the *regularity condition*[1]

$$a_{12} - c_{12} = d_{12} - b_{12} \tag{5}$$

is fulfilled, (4) is indeed identified with (3) by setting $r_{12} = \frac{1}{2}(a_{12} L_2 + L_2 a_{12}) - L_2 c_{12}$. Hence when the regularity condition (5) is fulfilled, the quantities $\operatorname{tr} L^k$ mutually Poisson-commute,

---

[1] The name 'regularity' will be motivated in section 5.

ensuring the integrability of the peakon models. The case $a = d$, $b = c = 0$ was first charac-terized by E. Sklyanin [6]; $a = d$, $b = c$ yields the so-called classical reflection algebra [3, 4].

We consider here the three integrable peakon equations discussed in e.g. [7] for which the key features of Poisson structure, integrability and Lax matrix, have been established:

1. The Camassa–Holm equation [8, 9]. Poisson structure for peakons is given in [10], Lax formulation in [11] although the Poisson structure here is not the one in [10]. We shall comment and relate the two structures in Section 2.

2. The Degasperis–Procesi equation [12, 13]. Poisson structure for peakons is given also in [10]. Lax formulation is given in [13], also commented in [14].

3. The Novikov equation [15]. Poisson structure for peakons is given in [16]. Lax formu-lation is given in [17].

Note that a fourth peakon-bearing integrable equation was identified (so-called modified Camassa–Holm equation [18, 19]), but peakon integrability properties are obstructed by the higher non-linearity of the modified Camassa–Holm equation, precluding the consistent re-duction of Poisson brackets and Hamiltonians to peakon variables [20].

We will establish in these three cases the existence of a quadratic $r$-matrix structure (4). In this paper, we only study the peakon dynamics inside the Weyl chambers defined by relative positions of the peakons $q_i$. We postpone the consideration of the possible crossings of the boundaries of the Weyl chambers to further studies.

We will show that the four parametrizing matrices $a$, $b$, $c$, $d$ are equal or closely connected to the Toda $A_n$ $r$-matrix [21]. This close connection can be understood in the Camassa–Holm and Novikov cases, via an identification between the Lax matrix of Camassa–Holm and the well-known Toda molecule Lax matrix [22]. In addition, the construction of the Novikov Lax matrix as $L^{\text{Nov}} = T L^{\text{CH}}$, where $T = \sum_{i,j}^{n} \big( 1 + \text{sgn}(q_i - q_j) \big) e_{ij}$, relates the two $r$-matrices by a twist structure. The occurence of $a_{12}$ in the Camassa–Holm context however also requires an understanding of the Camassa–Holm peakons Poisson bracket in [10] as a second Poisson bracket in the sense of Magri [23, 24], where the first Poisson bracket is the canonical structure $\{q_i, p_j\} = \delta_{ij}$, to be detailed in Section 2.

Each following section is now devoted to one particular model, resp. Camassa–Holm (Sec-tion 2), Degasperis–Procesi (Section 3), and Novikov (Section 4). We conclude with some comments and open questions. As a convention, to lighten the presentation, we will use the notation $a, b, c, d, \gamma$ in the Camassa–Holm case, $a', b', c', d', \gamma'$ for Degasperis–Procesi models, and $a'', b'', c'', d'', \gamma''$ for Novikov ones.

## 2 Camassa–Holm peakons

The Camassa–Holm shallow-water equation reads [8, 9]

$$u_t - u_{xxt} + 3uu_x = 2u_x u_{xx} + uu_{xxx} \,. \tag{6}$$

The $n$-peakon solutions take the form

$$u(x, t) = \sum_{i=1}^{n} p_i(t) \, e^{-|x - q_i(t)|} \,, \tag{7}$$

yielding a dynamical system for $p_i, q_i$:

$$\dot{q}_i = \sum_{j=1}^{n} p_j \, e^{-|q_i - q_j|} \,, \qquad \dot{p}_i = \sum_{j=1}^{n} p_i p_j \, \mathfrak{s}_{ij} \, e^{-|q_i - q_j|} \,, \tag{8}$$

where $\mathfrak{s}_{ij} = \text{sgn}(q_i - q_j)$. This discrete dynamical system is described by a Hamiltonian

$$H = \frac{1}{2} \sum_{i,j=1}^{n} p_i p_j \, e^{-|q_i - q_j|} \,, \tag{9}$$

such that

$$\dot{f} = \{f, H\} \,, \tag{10}$$

with the canonical Poisson structure:

$$\{p_i, p_j\} = \{q_i, q_j\} = 0 \,, \qquad \{q_i, p_j\} = \delta_{ij} \,. \tag{11}$$

The same dynamics is in fact also triggered [10] by the reduced Camassa–Holm Hamiltonian:

$$H = \sum_i p_i \,, \tag{12}$$

with the reduced Camassa–Holm Poisson structure (which is dynamical and "non-local"):

$$
\begin{aligned}
\{p_i, p_j\} &= \mathfrak{s}_{ij} \, p_i p_j e^{-|q_i - q_j|} \,, \\
\{q_i, p_j\} &= p_j e^{-|q_i - q_j|} \,, \\
\{q_i, q_j\} &= \mathfrak{s}_{ij} \left( 1 - e^{-|q_i - q_j|} \right) \,.
\end{aligned}
\tag{13}
$$

It is also encoded in the Lax formulation [8, 25]

$$\frac{dL}{dt} = [L, M] \,, \tag{14}$$

with

$$L = \sum_{i,j=1}^{n} L_{ij} e_{ij} \,, \qquad L_{ij} = \sqrt{p_i p_j} \, e^{-\frac{1}{2}|q_i - q_j|} \,, \tag{15}$$

where $e_{ij}$ is the $n \times n$ elementary matrix with 1 at position $(i, j)$ and 0 elsewhere.

## 2.1 The linear Poisson structure

We summarize here the results obtained in [11]. The Poisson structure (11) endows the Lax matrix (15) with a linear $r$-matrix structure

$$\{L_1, L_2\} = [r_{12}, L_1] - [r_{21}, L_2] \,, \quad \text{with} \quad r_{12} = a_{12} - b_{12} \,. \tag{16}$$

In (16), $a_{12}$ is the $A_{n-1}$ Toda $r$-matrix

$$a_{12} = \frac{1}{4} \sum_{i,j=1}^{n} \mathfrak{s}_{ij} \, e_{ij} \otimes e_{ji} = -a_{21} \qquad \text{and} \qquad b_{12} = -a_{12}^{t_2} \,, \tag{17}$$

with by convention $\text{sgn}(0) = 0$ and $t_2$ denotes the transposition in space 2. Connection of the Lax matrix with the Toda $r$-matrix structure was already pointed out in [11]. The $r$-matrix structure (16) is indeed identified with the same structure occuring in the so-called Toda lattice models [24]. One can add that the $r$-matrix structure for the Toda lattice in [11] and the peakon dynamics in (16) is directly identified with the well-known $r$-matrix structure for Toda molecule models [22] (strictly speaking, this identification holds in the first Weyl chamber, where $\text{sgn}(q_i - q_j) = \text{sgn}(i - j)$). Indeed, both Toda lattice and peakon Lax matrices

endowed with the canonical Poisson structure (11) are representations of the abstract $A_{n-1}$ Toda molecule structure

$$L = \sum_i x_i h_i + \tfrac{1}{2} \sum_{\alpha \in \Delta_+} x_\alpha (e_\alpha + e_{-\alpha}), \tag{18}$$

where $\{x_\alpha, x_\beta\} = x_{\alpha+\beta}$ and $\{x_i, x_\alpha\} = \alpha(i) x_\alpha$, with $[h_i, e_\alpha] = \alpha(i) e_\alpha$. Here, $h_i$ and $e_\alpha$ denote the usual Cartan elements and root vectors with root $\alpha$, $\Delta_+$ denotes the positive root system, and the Killing form is normalized as $\text{tr}(h_i h_j) = \delta_{ij}$ and $\text{tr}(e_\alpha e_\beta) = \delta_{\alpha\beta}$.

In the highly degenerate case of Toda Lax matrix (where $x_\alpha = 0$ for non-simple roots $\alpha$), it is directly checked that $a$ and $b$ yield the same contribution to (16), implying that the Toda Lax matrix has an $r$-matrix structure parametrized by $a_{12}$ solely, as it is well-known [21].

## 2.2 The quadratic Poisson structure

The new result which we shall elaborate on now is stated as:

**Proposition 2.1** *The Poisson structure* (13) *endows the Lax matrix* (15) *with a quadratic $r$-matrix structure:*

$$\{L_1, L_2\} = [a_{12}, L_1 L_2] - L_2 b_{12} L_1 + L_1 b_{12} L_2, \tag{19}$$

*where $a_{12}$ and $b_{12}$ are given in* (17).

**Proof:** Direct check by computing the Poisson bracket $\{L_{ij}, L_{kl}\}$ on the left hand side and right hand side. The antisymmetry of the Poisson structure, explicitly realized by (19), allows to eliminate "mirror display", i.e. $(ij, kl) \leftrightarrow (kl, ij)$. The invariance of the Poisson structure (19) under each operation $t_1$ and $t_2$ is due to the symmetry $L^t = L$ of (15), the identification of $b_{12} = -a_{12}^{t_2}$, and the antisymmetry $a_{12}^{t_1 t_2} = -a_{12}$. It allows to eliminate transposed displays $(ij, kl) \leftrightarrow (ji, kl) \leftrightarrow (ji, lk) \leftrightarrow (ij, kl)$ and to check only a limited number of cases (indeed 13 cases). ∎

Remark that the form (19) ensures that the regularity condition (5) is trivially obeyed. The Poisson bracket structure (16) is indeed a restriction to a coadjoint orbit, of the well-known Kostant–Kirillov bracket on $sl(n)$ restricted to symmetric matrices (on the full set of which it does not determine a symplectic structure). The quadratic Poisson bracket (19) defined from the antisymmetric and symmetric components $a$, $b$ of the linear structure, is a related natural object (also known as Sklyanin bracket). We expect that its intrinsic interpretation as a Poisson structure on symmetric matrices may be connected to the notion of Heisenberg double introduced in [2].

The Poisson structure (13), identified as a second Poisson structure in the sense of Magri [23], yields the natural quadratization (19) of the $r$-matrix structure (16). This fact is consistent with the fact that (13) is obtained by reduction to peakon variables of the second Poisson structure of Camassa–Holm, built in [10], while (11) is obtained by reduction of the first Camassa–Holm Poisson structure. Reduction procedure (from fields to peakon variables) and recursion construction (*à la* Magri, see [24]) are therefore compatible in this case, and the compatibility extends to the $r$-matrix structures of the reduced variables. Such a consistency at the $r$-matrix level is not an absolute rule. For instance, the first and second Poisson structures for the Calogero–Moser model yield $r$-matrix structures, "linear" [26] and "quadratic" [27], but with different $r$-matrices.

## 2.3 The Yang–Baxter relations

**Quadratic structure.** As is known from general principles [5], quadratic Poisson $r$-matrix structures obey consistency quadratic equations of Yang–Baxter type to ensure Jacobi identity of Poisson brackets. In the case of original Camassa–Holm pair $(a, b)$ in (17), the skew-symmetric element $a_{12}$ obeys the modified Yang–Baxter equation:

$$[a_{12}, a_{13}] + [a_{12}, a_{23}] + [a_{13}, a_{23}] = \frac{1}{16}\left(\Omega_{123} - \Omega_{123}^{t_1 t_2 t_3}\right), \quad \text{with } \Omega_{123} = \sum_{i,j,k=1}^{n} e_{ij} \otimes e_{jk} \otimes e_{ki}. \quad (20)$$

The symmetric element $b_{12}$ obeys an adjoint-modified Yang–Baxter equation directly obtained from transposing (20) over space 3:

$$[a_{12}, b_{13}] + [a_{12}, b_{23}] + [b_{13}, b_{23}] = \frac{1}{16}\left(-\Omega_{123}^{t_3} + \Omega_{123}^{t_1 t_2}\right). \quad (21)$$

Cancellation of a suitable combination of (20) and (21) with all permutations added, together with symmetry of the Camassa–Holm Lax matrix, allows to then check explicitly Jacobi identity for $L^{CH}$ and Poisson structure (19).

**Linear structure.** The Jacobi identity for the linear Poisson structure (16) also follows from (20) and (21). It is equivalent to the cyclic relation:

$$[[r_{12}, r_{13}] + [r_{12}, r_{23}] + [r_{32}, r_{13}], L_1] + cyclic = 0, \quad (22)$$

where *cyclic* stands for sum over cyclic permutations of $(1, 2, 3)$.

The Yang–Baxter "kernel" $[r_{12}, r_{13}] + [r_{12}, r_{23}] + [r_{32}, r_{13}]$ must now be evaluated. In many models, it is known to be equal to 0 (classical Yang–Baxter equation) or to a combination of the cubic Casimir operators $\Omega_{123}$ and $\Omega_{123}^{t_1 t_2 t_3}$ (modified Yang–Baxter equation). If any of these two sufficient conditions holds, (22) is then trivial. However, in the Camassa–Holm case, the situation is more involved. Indeed from (20) and (21), and denoting the Casimir term $C_{123} \equiv \Omega_{123} - \Omega_{123}^{t_1 t_2 t_3}$, one has:

$$[r_{12}, r_{13}] + [r_{12}, r_{23}] + [r_{32}, r_{13}] = C_{123} + C_{123}^{t_3} + C_{123}^{t_2} - C_{123}^{t_1}, \quad (23)$$

which is neither a cubic Casimir nor even cyclically symmetric. Realization of (22) indeed follows from explicit direct cancellation of the first (factorizing) Casimir term in (23) under commutation with $L_1 + L_2 + L_3$ and cross-cancellation of the remaining 9 terms, using in addition the invariance of $L$ under transposition. We have here a textbook example of an $r$-matrix parametrizing a Poisson structure for a Lax matrix without obeying one of the canonical classical Yang–Baxter equations.

## 3 Degasperis–Procesi peakons

This integrable shallow-water equation reads [13]

$$u_t - u_{xxt} + 4uu_x = 3u_x u_{xx} + uu_{xxx}. \quad (24)$$

Note that, together with the Camassa–Holm equation, it is a particular case of the so-called $b$-equations:

$$u_t - u_{xxt} + (\beta + 1)uu_x = \beta u_x u_{xx} + uu_{xxx}, \quad (25)$$

for which integrability properties are established for $\beta = 2$ (Camassa–Holm) and $\beta = 3$ (Degasperis–Procesi), by an asymptotic integrability approach [13]. This approach fails at $\beta = 4$.

### 3.1 The quadratic Poisson structure

For $\beta = 3$, $n$-peakon solutions are parametrized as

$$u(x,t) = \frac{1}{2} \sum_{j=1}^{n} p_j(t) \, e^{-|x-q_j(t)|}, \tag{26}$$

yielding a dynamical system:

$$\dot{p}_j = 2 \sum_{k=1}^{n} p_j p_k \, \mathfrak{s}_{jk} \, e^{-|q_j-q_k|},$$

$$\dot{q}_j = \sum_{k=1}^{n} p_k \, e^{-|q_j-q_k|}. \tag{27}$$

Note the extra factor 2 in $\dot{p}_j$ compared with the Camassa–Holm equation.
The Lax matrix is now given by

$$L_{ij} = \sqrt{p_i p_j} \left( T_{ij} - \mathfrak{s}_{ij} \, e^{-|q_i-q_j|} \right), \tag{28}$$

with

$$T_{ij} = 1 + \mathfrak{s}_{ij}, \quad i,j = 1,\ldots,n. \tag{29}$$

The dynamical equations (27) derive from the Hamiltonian $H = \text{tr}(L)$ and the Poisson structure obtained by reduction from the canonical Poisson structure of Degasperis–Procesi:

$$\{p_i, p_j\} = 2 p_i p_j \mathfrak{s}_{ij} \, e^{-|q_i-q_j|},$$

$$\{q_i, p_j\} = p_j e^{-|q_i-q_j|}, \tag{30}$$

$$\{q_i, q_j\} = \frac{1}{2} \mathfrak{s}_{ij} \left( 1 - e^{-|q_i-q_j|} \right).$$

Again one notes the non-trivial normalization of the Poisson brackets in (30) compared with (13) which will have a very significant effect on the $r$-matrix issues. Let us note that the Hamiltonian associated to a time evolution $\dot{f} = \{f, H\}$ consistent with the dynamics (27) is in fact the conserved quantity noted $P$ in [13]. The Hamiltonian $H$ in [13] is $\text{tr} \, L^2$.

Let us now state the key result of this section.

**Proposition 3.1** *The Poisson structure* (30) *endows the Lax matrix $L$ given in* (28) *with a quadratic $r$-matrix structure:*

$$\{L_1, L_2\} = [a'_{12}, L_1 L_2] - L_2 b'_{12} L_1 + L_1 b'_{12} L_2, \tag{31}$$

*where (we remind our convention that* $\text{sgn}(0) = 0$*)*

$$a'_{12} = \frac{1}{2} \sum_{i,j} \mathfrak{s}_{ij} \, e_{ij} \otimes e_{ji} = 2 \, a_{12}, \tag{32}$$

$$b'_{12} = -\frac{1}{2} \sum_{i,j} \mathfrak{s}_{ij} \, e_{ij} \otimes e_{ij} - \frac{1}{2} \mathcal{Q}_{12}, \quad \text{with} \quad \mathcal{Q}_{12} = \sum_{i,j=1}^{n} e_{ij} \otimes e_{ij}. \tag{33}$$

**Proof:** : By direct check of the left hand side and right hand side of (31). Since the Lax matrix $L$ is neither symmetric nor antisymmetric, many more cases of inequivalent index displays occur. More precisely, 12 four-indices, 18 three-indices and 5 two-indices must be checked. ∎

The regularity condition (5) is again trivially fulfilled.

## 3.2 The Yang–Baxter equations

The matrix $a'_{12}$ in the Degasperis–Procesi model is essentially the linear Toda $r$-matrix as in the Camassa–Holm model. On the contrary, the $b'$ component of the quadratic structure (31) must differ from the $b$ component in the quadratic Camassa–Holm bracket (19) by an extra term proportional to $\mathcal{Q}_{12} = \mathcal{P}^{t_1}_{12} = \mathcal{P}^{t_2}_{12} = \mathcal{Q}_{21}$, where $\mathcal{P}_{12} = \sum_{i,j=1}^{n} e_{ij} \otimes e_{ji}$ is the permutation operator between space 1 and space 2. For future use, we also note the property

$$\mathcal{P}^2_{12} = \mathbb{I}_n \otimes \mathbb{I}_n \quad \text{and} \quad \mathcal{P}_{12} M_1 M'_2 \mathcal{P}_{12} = M'_1 M_2, \tag{34}$$

$$M_1 \mathcal{Q}_{12} = M^t_2 \mathcal{Q}_{12} \quad \text{and} \quad \mathcal{Q}_{12} M_1 = \mathcal{Q}_{12} M^t_2, \tag{35}$$

which holds for any $n \times n$ matrices $M$ and $M'$.

**Remark 3.1** *Since the Lax matrix $L^{CH}$ of Camassa–Holm peakons is a symmetric c-number matrix, $L = L^t$, one checks that*

$$\mathcal{P}_{12} L_1 L_2 = L_2 L_1 \mathcal{P}_{12} \quad \text{and} \quad L_1 \mathcal{P}^{t_1}_{12} L_2 = L_2 \mathcal{P}^{t_1}_{12} L_1. \tag{36}$$

*Hence the $(a', b')$ pair of r-matrices yielding the quadratic Poisson structure for Degasperis–Procesi peakons yields (up to a factor 2) the quadratic Poisson structure for Camassa–Holm peakons with a pair $(a, b - \frac{1}{4}\mathcal{Q})$, since the extra contribution $-L_2 \mathcal{Q}_{12} L_1 + L_1 \mathcal{Q}_{12} L_2$ cancels out. In this case, we will call this pair an alternative presentation for Camassa–Holm peakons.*

The Degasperis–Procesi $(a', b')$ pair obeys a set of classical Yang–Baxter equations which is simpler than the Camassa–Holm pair $(a, b)$. The skew-symmetric element $a_{12}$ still obeys a modified Yang–Baxter equation

$$[a'_{12}, a'_{13}] + [a'_{12}, a'_{23}] + [a'_{13}, a'_{23}] = \frac{1}{4}\left(\Omega_{123} - \Omega^{t_1 t_2 t_3}_{123}\right), \tag{37}$$

but the symmetric element $b'$ obeys an adjoint-modified Yang–Baxter equation with zero right-hand-side:

$$[a'_{12}, b'_{13}] + [a'_{12}, b'_{23}] + [b'_{13}, b'_{23}] = 0. \tag{38}$$

**Remark 3.2** *A term proportional to $\mathcal{P}_{12}$ can be added to $a'_{12}$, leading to a matrix $\widetilde{a}'_{12} = a'_{12} + \frac{1}{2}\mathcal{P}_{12}$. This term is optional, it does not change the Poisson brackets, nor the regularity condition. If added, it allows the relation $b'_{12} = -(\widetilde{a}'_{12})^{t_2}$, which already occurred for the Camassa–Holm model. However, such a term breaks the antisymmetry relation $a'_{21} = -a'_{12}$, which has deep consequences at the level of Yang–Baxter equations. Indeed, the form of the left-hand-side in (20) heavily relies on this antisymmetry property of $a'_{12}$. In fact, if one computes "naively" $[\widetilde{a}'_{12}, \widetilde{a}'_{13}] + [\widetilde{a}'_{12}, \widetilde{a}'_{23}] + [\widetilde{a}'_{13}, \widetilde{a}'_{23}]$, one finds exactly zero and could be tempted to associate it to a Yang–Baxter equation with zero right-hand-side. Yet, the "genuine" Yang–Baxter equation, i.e. the relation ensuring the associativity of the Poisson brackets, plugs the $\Omega$-term back into the game, leading in fine to again a modified Yang–Baxter equation.*

## 3.3 Search for a linear Poisson structure

Contrary to the Camassa–Holm peakon case, the canonical Poisson bracket (11) is not compatible with the soliton-derived Poisson bracket structure (30). Indeed, the linear pencil $\{\cdot, \cdot\}_{\text{can}} + \lambda\{\cdot, \cdot\}_{DP}$ ("can" is for canonical and DP for Degasperis–Procesi) does not obey Jacobi identity due to extra non-cancelling contributions from the non-trivially *scaled* brackets

of $\{p,p\}$ and $\{q,q\}$ in (30). The statement is consistent with the fact, pointed out in [10], that no second local Poisson structure exists in the Degasperis–Procesi case, contrary to the Camassa–Holm case (where it is denoted as $B_1$ in [10]).

Consistently with the absence of a second local Poisson structure for soliton Degasperis–Procesi equation yielding a "linear" $r$-matrix structure for the Lax matrix, one observes that the associated linear $r$-matrix structure naively defined by $r_{12} = a'_{12} + b'_{12}$, does not yield consistent Poisson brackets for the variables in the Lax matrix. If one indeed sets

$$\{L_1, L_2\} = [a'_{12} + b'_{12}, L_1] - [a'_{21} + b'_{21}, L_2], \tag{39}$$

the Poisson brackets for individual coordinates of $L$ are inconsistent, due to the antisymmetric part in (28), contrary to the Camassa–Holm case where $L_{ij} = L_{ji}$.

We also checked using software calculations that (at least for $n$ running from 2 to 7) that there is no non-trivial linear combination $r'_{12} = x\, a'_{12} + y\, b'_{12}$ such that the relation $\{L_1, L_2\} = [r'_{12}, L_1] - [r'_{21}, L_2]$ with $L$ given in (28), yield a consistent Poisson structure for the $(p_i, q_j)$ variables.

The peakon Lax matrix (28) realizes therefore an interesting example of a non-dynamical quadratic $(a', b')$ Poisson structure where there is no associated linear $r$-matrix structure. The exact form, or even the existence, of such linear $r$-matrix structure for Degasperis–Procesi peakons remains an open question.

## 4 Novikov peakons

The Novikov shallow-wave equation reads

$$u_t - u_{xxt} + 4u^2 u_x = 3uu_x u_{xx} + u^2 u_{xxx}, \tag{40}$$

showing now a cubic non-linearity instead of a quadratic one as in Camassa–Holm or Degasperis–Procesi. Originally proposed by Novikov [28] as an integrable partial differential equation, it was later shown [16] to have integrable peakons:

$$u(x,t) = \sum_{i=1}^{n} p_i(t)\, e^{-|x-q_i(t)|}. \tag{41}$$

### 4.1 The quadratic Poisson structure

The complete integrability structure was established in [17]. The dynamical system for $p_i, q_i$ reads

$$\dot{p}_i = p_i \sum_{j,k=1}^{n} \mathfrak{s}_{ij}\, p_j p_k\, e^{-|q_i-q_j|-|q_i-q_k|},$$

$$\dot{q}_i = \sum_{j,k=1}^{n} p_j p_k\, e^{-|q_i-q_j|-|q_i-q_k|}, \tag{42}$$

still with the notation $\mathfrak{s}_{ij} = \mathrm{sgn}(q_i - q_j)$. They constitute a Hamiltonian system where the Poisson structure takes the following form:

$$\begin{aligned}
\{p_i, p_j\} &= \mathfrak{s}_{ij}\, p_i p_j\, e^{-2|q_i-q_j|}, \\
\{q_i, p_j\} &= p_j\, e^{-2|q_i-q_j|}, \\
\{q_i, q_j\} &= \mathfrak{s}_{ij}\left(1 - e^{-2|q_i-q_j|}\right).
\end{aligned} \tag{43}$$

The conserved Hamiltonians are obtained as traces of a Lax matrix

$$L = TPEP, \tag{44}$$

where

$$T_{ij} = 1 + \mathfrak{s}_{ij}, \qquad P_{ij} = p_i \delta_{ij}, \qquad E_{ij} = e^{-|q_i - q_j|}. \tag{45}$$

In other words, the time evolution (42) is described by the Hamilton equation $\dot{f} = \{f, H\}$, with $H = \frac{1}{2}\operatorname{tr} L$ and the PB (43).

Redefining now

$$\bar{q}_j = 2q_j \qquad \text{and} \qquad \bar{p}_j = p_j^2, \tag{46}$$

yield a Poisson structure

$$
\begin{aligned}
\{\bar{p}_i, \bar{p}_j\} &= 4\mathfrak{s}_{ij}\,\bar{p}_i\bar{p}_j e^{-|\bar{q}_i - \bar{q}_j|}, \\
\{\bar{q}_i, \bar{p}_j\} &= 4\bar{p}_j e^{-|\bar{q}_i - \bar{q}_j|}, \\
\{\bar{q}_i, \bar{q}_j\} &= 4\mathfrak{s}_{ij}\left(1 - e^{-|\bar{q}_i - \bar{q}_j|}\right),
\end{aligned}
\tag{47}
$$

identical to the Camassa–Holm peakon structure (13) up to a factor 4. The Lax matrix now reads

$$L_{ij} = \sum_{k=1}^{n} T_{ik} \sqrt{\bar{p}_k \bar{p}_j}\, e^{-\frac{1}{2}|\bar{q}_j - \bar{q}_k|}, \tag{48}$$

exactly identified with $TL^{CH}$.

Hence, the Novikov peakons are in fact described by a Lax matrix simply twisted from the Camassa–Holm Lax matrix ($L \rightarrow TL$) and an identical Poisson bracket, a fact seemingly overlooked in [17]. The $r$-matrix structure immediately follows, but several inequivalent structures are identified due to the gauge covariance pointed out in section 3:

**Proposition 4.1** *The Poisson structure* (43) *endows the Lax matrix* (48) *with a set of quadratic r-matrix structure*

$$\{L_1, L_2\} = a''_{12}L_1L_2 - L_1L_2d''_{12} + L_1b''_{12}L_2 - L_2c''_{12}L_1, \tag{49}$$

*where*

$$
\begin{aligned}
a''_{12} &= 4\,T_1 T_2\, a_{12}\, T_1^{-1} T_2^{-1}, & d''_{12} &= 4\,a_{12}, \\
b''_{12} &= T_2\left(-4a_{12}^{t_2} - \mathcal{Q}_{12}\right)T_2^{-1}, & c''_{12} &= b''_{21},
\end{aligned}
\tag{50}
$$

*and $a_{12}$ is given in* (17).

The proof follows trivially from section 2 and gauge invariance in section 3.

The regularity condition (5) is fulfilled by this Poisson structure. Although less trivial than in the previous two cases this property will be proved in the next section.

## 4.2 The Yang–Baxter equations

The Yang–Baxter equations for (50) follow immediately by suitable conjugations by $T$ of the Yang–Baxter equations for the alternative form of Degasperis–Procesi structures matrices. Precisely, from the redefinitions in (50) the Yang–Baxter equations for $a''$, $b''$, $c''$ and $d''$ read

$$
\begin{aligned}
[a''_{12}, a''_{13}] + [a''_{12}, a''_{23}] + [a''_{13}, a''_{23}] &= \Omega_{123} - \Omega_{123}^{t_1 t_2 t_3}, \\
[d''_{12}, d''_{13}] + [d''_{12}, d''_{23}] + [d''_{13}, d''_{23}] &= \Omega_{123} - \Omega_{123}^{t_1 t_2 t_3}, \\
[a''_{12}, b''_{13}] + [a''_{12}, b''_{23}] + [b''_{13}, b''_{23}] &= 0, \\
[d''_{12}, c''_{13}] + [d''_{12}, c''_{23}] + [c''_{13}, c''_{23}] &= 0,
\end{aligned}
\tag{51}
$$

where in writing the right-hand-side of the relation for $a''$, we have used the property that $\Omega_{123} - \Omega_{123}^{t_1 t_2 t_3}$ commutes with any product of the form $M_1 M_2 M_3$ for any matrix $M$ ($M = T$ for the present calculation). Note that the adjoint Yang–Baxter equations for $b''$ and $c''$ remain with zero right-hand-side, despite the conjugations by $T$ depend on the matrices (e.g. $a''$ or $b''$) one considers.

**Linear structure.**  As for the Degasperis–Procesi case, we looked for a linear combination $r''_{12} = x\, a''_{12} + y\, b''_{12} + z\, c''_{12} + t\, d''_{12}$ such that the Poisson structure $\{L_1, L_2\} = [r''_{12}, L_1] - [r''_{21}, L_2]$ with $L$ given in (48), yield a consistent Poisson structure for the $(p_i, q_j)$ variables. For $n > 2$, the only solution is given by $r''_{12} = x\,(a''_{12} + b''_{12} - c''_{12} - d''_{12})$ which is identically zero due to the regularity relation (5). The calculation was done using a symbolic computation software for $n$ running from 3 to 5. Hence, in the generic case, we conjecture that there is no non-trivial linear structure, at least directly associated to the quadratic one. Again, the existence (and the exact form) of such linear $r$-matrix structure for general Novikov peakons remains an open question.

In the particular case $n = 2$, there is indeed a solution related to the solution

$$
r''_{12} = \frac{1}{2}\Big( a''_{12} - b''_{12} - c''_{12} + d''_{12} \Big) = a''_{12} - c''_{12} = \begin{pmatrix} 1 & 0 & 0 & 0 \\ 2 & 0 & -1 & 0 \\ 0 & 1 & 0 & 0 \\ 2 & 0 & -2 & 1 \end{pmatrix}.
\tag{52}
$$

The $r$-matrix obeys the modified Yang–Baxter relation

$$
[r''_{12}, r''_{13}] + [r''_{12}, r''_{23}] + [r''_{32}, r''_{13}] = \Omega_{123} - \Omega_{123}^{t_1 t_2 t_3},
\tag{53}
$$

and leads to PB of the form

$$
\{\bar{p}_1, \bar{p}_2\} = -4\, \mathfrak{s}_{12} \sqrt{\bar{p}_1 \bar{p}_2}\, e^{-\frac{1}{2}|\bar{q}_1 - \bar{q}_2|},
$$

$$
\{\bar{q}_1 - \bar{q}_2, \bar{p}_1\} = 4 \sqrt{\frac{\bar{p}_1}{\bar{p}_2}}\, e^{-\frac{1}{2}|\bar{q}_1 - \bar{q}_2|} + 4,
\tag{54}
$$

$$
\{\bar{q}_1 - \bar{q}_2, \bar{p}_2\} = -4 \sqrt{\frac{\bar{p}_2}{\bar{p}_1}}\, e^{-\frac{1}{2}|\bar{q}_1 - \bar{q}_2|} - 4.
$$

Indeed since the combination $\bar{q}_1 + \bar{q}_2$ does not appear in the expression of $L$, one can realize a consistent associative, albeit degenerate, Poisson bracket by setting $\{\bar{q}_1 + \bar{q}_2, X\} = 0$ for all $X$. In order to obtain a non-degenerate Poisson structure, one has to implement the PBs $\{\bar{q}_1 + \bar{q}_2, X\} = f_X(\bar{p}_1, \bar{p}_2, \bar{q}_1, \bar{q}_2)$ for $X = \bar{p}_1, \bar{p}_2, \bar{q}_1 - \bar{q}_2$ and solve all functional equations (on the functions $f_X$) provided by the Jacobi identities. Finally, among the solutions, find a solution that leads to a non-degenerate Poisson structure. This is beyond the scope of this article.

## 4.3  Dual presentation for Novikov peakons

Let us remark that the form of the Novikov Lax matrix $L^N = T\, L^{CH}$ suggests a dual presentation for the Novikov peakons. Indeed, one can introduce the Lax matrix $\widetilde{L}^N = L^{CH}\, T$, which takes explicitly the form

$$
\widetilde{L}_{ij} = \sum_{k=1}^{n} \sqrt{\bar{p}_k \bar{p}_i}\, e^{-\frac{1}{2}|\bar{q}_i - \bar{q}_k|}\, T_{kj}.
\tag{55}
$$

In that case, the PB (43) have still a quadratic structure of the form (49), but with now

$$
\widetilde{a}''_{12} = 4\, a_{12}, \qquad\qquad \widetilde{d}''_{12} = 4\, T_1^{-1} T_2^{-1} a_{12} T_1 T_2,
$$

$$
\widetilde{b}''_{12} = T_1^{-1}\big( -4 a_{12}^{t_2} - \mathcal{Q}_{12} \big) T_1, \qquad \widetilde{c}''_{12} = \widetilde{b}''_{21}.
\tag{56}
$$

It is easy to see that these matrices still obey precisely the same Yang–Baxter relations (51).

The Hamiltonians one constructs using $\widetilde{L}^N$ are exactly the same as for $L^N$. Thus, we get a dual presentation of exactly the same model and the same Hamiltonians. This property extends to the calculation presented in the next section.

# 5  Non-trivial boundary terms for peakons

A general construction of Poisson-commuting Hamiltonians from any quadratic structure (4) is based on the product of two dynamical objects: $L$ obeying (4) and $Q$ obeying a dual Poisson structure:

$$\{Q_1, Q_2\} = Q_1 Q_2 a_{12} - d_{12} Q_1 Q_2 + Q_2 b_{12} Q_1 - Q_1 c_{12} Q_2\,, \tag{57}$$

with $\{Q, L\} = 0$. It is then easy to show that $QL$ obeys a linearizable (albeit dynamical) $r$-matrix Poisson structure:

$$\{Q_1 L_1, Q_2 L_2\} = [r_{12}, Q_1 L_1] - [r_{21}, Q_2 L_2]\,, \tag{58}$$

where

$$r_{12} = \tfrac{1}{2}(d_{12} Q_2 L_2 + Q_2 L_2 d_{12}) - Q_2 b_{12} L_2\,. \tag{59}$$

The traces $\operatorname{tr}\left((QL)^k\right)$ are thus seen to be Poisson-commuting dynamical quantities. Following the traditional approach of integrable systems with boundary [6, 29], these boundary matrices characterize the boundary condition. Note however that this approach is mainly used in the quantum case: in the classical (i.e. PB) case, the relation is less clear and deserves a more detailed study.

We restrict ourselves in this paper to the case when the dual term $Q$ is a non dynamical matrix $\gamma$, hence obeys the purely algebraic, dual reflection equation:

$$\gamma_1 \gamma_2 a_{12} - d_{12} \gamma_1 \gamma_2 + \gamma_2 b_{12} \gamma_1 - \gamma_1 c_{12} \gamma_2 = 0\,. \tag{60}$$

More general peakon Hamiltonians can then be defined for each solution $\gamma$ of the dual classical reflection equation (60). In the case where $\gamma$ is diagonal, one in fact re-obtains the original Hamiltonians by a rescaling of the $p_i$ variables, see below. The Hamiltonians take the form $\operatorname{tr}\left((\gamma L)^k\right)$. The solution $\gamma = \mathbb{I}_n$ exists whenever the condition (5) holds, motivating its designation as *regularity condition*. Note that in the Freidel–Maillet approach [5], the immediate correspondence between solutions $\gamma$ of the dual equation and solutions $\gamma^{-1}$ of the direct equation (r.h.s. of (4) = 0) is used to yield an equivalent form of the commuting Hamiltonians.

We shall now propose classes of invertible solutions to the classical reflection equation (60) for each of the three peakon cases. Let us first emphasize that any solution $\gamma'$ of (60) where $a'_{12} = 2a_{12}$, $b'_{12}$, $c'_{12} = b'_{21}$, $d'_{12} = 2a_{12}$ are the matrices associated with the Degasperis–Procesi peakons, or to the alternative presentation of Camassa–Holm peakons, corresponds to a solution $\gamma'' = \gamma' T^{-1}$ for the reflection equation associated to the Novikov peakons, since the structure matrices are related as: $b''_{12} = 2T_2 b'_{12} T_2^{-1}$, $c''_{12} = b''_{21}$, $a''_{12} = 2T_1 T_2 a'_{12} T_1^{-1} T_2^{-1}$ and $d''_{12} = 2a'_{12}$. Hence the Degasperis–Procesi case provides solutions for the two other peakon models.

**Lemma 5.1** *If $\gamma'$ is a solution of the Degasperis–Procesi reflection equation* (60)*, then for any diagonal matrix $D$, the matrix $D\gamma'D$ is also a solution of the reflection equation.*

**Proof:** The structure matrices $a'_{12}$ and $b'_{12}$ of (60) for the Degasperis–Procesi peakons obey :

$$a'_{12} D_1 D_2 = D_1 D_2 a'_{12} \qquad \text{and} \qquad D_1 b'_{12} D_2 = D_2 b'_{12} D_1\,, \tag{61}$$

where we used also the property (35). Now, multiplying (60) by $D_1 D_2$ on the right or / and the left hand sides, and using the above properties of $a'_{12}$ and $b'_{12}$, leads to the desired result. ∎

Note that the transformation $\gamma' \to D\gamma'D$ is equivalent to a canonical redefinition $p_i \to d_i^2 p_i$.

**Proposition 5.2** *For Degasperis–Procesi peakons, and for n arbitrary, we have two fundamental solutions:*

> *(i) the unit matrix $\mathbb{I}_n$,*
> *(ii) the matrix T introduced in* (29).

*Moreover, when n is even, we have an additional solution, whose explicit form depends on the Weyl chamber we consider for the variables $q_i$. In the first Weyl chamber, where $q_i > q_j \iff i > j$, it takes the form*

$$(iii) \qquad S^{id} = \sum_{i=0}^{\frac{n}{2}-1} \left( e_{2i+1,2i+2} - e_{2i+2,2i+1} \right) = \mathbb{I}_{n/2} \otimes \begin{pmatrix} 0 & 1 \\ -1 & 0 \end{pmatrix}. \tag{62}$$

*In any other Weyl chamber defined by a permutation $\sigma$ such that $q_i > q_j \iff \sigma(i) > \sigma(j)$, the solution $S^\sigma$ takes the form*

$$S^\sigma = \sum_{i=0}^{\frac{n}{2}-1} \left( e_{\sigma(2i+1),\sigma(2i+2)} - e_{\sigma(2i+2),\sigma(2i+1)} \right). \tag{63}$$

> *Using the lemma 5.1, it leads to 2 (resp. 3) classes of solutions for n odd (resp. even).*
>
> *All these solutions are also valid for the alternative presentation of Camassa–Holm, and (once multiplied on the left by T) for the Novikov model. The solutions (i) and (iii) are also valid for the original Camassa–Holm model.*

**Proof:** ($i$) The unit matrix is trivially a solution since (60) is then the regularity condition.
($ii$) The reflection equation for $T$ projected on a generic element $e_{ij} \otimes e_{kl}$ contains explicitly the indices $i, j, k, l$, and possibly two summation indices corresponding to the products by $\gamma'_1$ and $\gamma'_2$. Since the entries of the matrices depend on the indices only through the sign function $\mathrm{sgn}(r-s)$, it is sufficient to check the relations for small values of $n$. We verified them through a symbolic calculation software for $n$ running from 2 to 8.
($iii$) Similarly, the reflection equation for $S$ needs to be checked for small values of $n$. We verified it through a symbolic calculation software for $n$ running from 2 to 8. ∎

**Remark 5.1** *For all three models, if $\gamma$ is a solution to the reflection equation, then $\gamma^t$ is also a solution. For Camassa–Holm peakons, $\gamma^{-1}$ is also a solution. However, the classes of solutions induced by these transformations falls in the ones already presented in proposition 5.2.*

Note that dressing $T$ by the diagonal matrix $D = \mathrm{diag}((-1)^i)$ yields $T^{-1}$, which is also a solution to the Degasperis–Procesi reflection equation. Moreover, it proves that the unit matrix is also a solution of the reflection equation (60) in the Novikov case, proving the property mentioned in the previous section that regularity condition is fulfilled by the Novikov $r$-matrix structure.

# 6 Hamiltonians

Now that the quadratic $r$-matrix structure have been defined, we are in position to compute higher Hamiltonians for each of the three classes of peakons. These Hamiltonians will be PB-commuting, with the Poisson brackets (13), (30) or (43), depending on the peakon model that

is to say the Lax matrices (15), (28) or (48). We provide also some cases of Hamiltonians with non-trivial boundary terms. Note however that when non diagonal boundary terms enter into the game, the square root of the momenta $p_j$ may be involved, notably for boundary terms associated to the $T$ matrix, see e.g. (66). In that case, one should check the sign of these momenta, and the corresponding conserved quantity will be real only in the domains where the positivity of $p_j$ is preserved.

## 6.1 Camassa–Holm Hamiltonians

In addition to the peakon Hamiltonian $H_{CH} = \operatorname{tr} L = \sum_i p_i$, we get for instance

$$
\begin{aligned}
H_{CH}^{(1)} &= \operatorname{tr} L = \sum_i p_i\,, \\
H_{CH}^{(2)} &= \operatorname{tr} L^2 = \sum_{i,j} p_i p_j\, e^{-|q_i-q_j|}\,, \\
H_{CH}^{(3)} &= \operatorname{tr} L^3 = \sum_{i,j,k} p_i p_j p_k\, e^{-\frac{1}{2}|q_i-q_j|}\, e^{-\frac{1}{2}|q_j-q_k|}\, e^{-\frac{1}{2}|q_k-q_i|}\,.
\end{aligned}
\tag{64}
$$

We recognize in $H_{CH}^{(1)}$ and $H_{CH}^{(2)}$ the usual Camassa–Holm Hamiltonians, as computed e.g. in [10].

**Diagonal boundary term.** If one chooses $\gamma = D$ as a diagonal solution to the reflection equation, we get another series of PB-commuting Hamiltonians:

$$
\begin{aligned}
\operatorname{tr}(DL) &= \sum_i d_i p_i\,, \\
\operatorname{tr}\big((DL)^2\big) &= \sum_i d_i^2 p_i^2 + 2\sum_{i<j} d_i d_j p_i p_j\, e^{-|q_i-q_j|}\,, \\
\operatorname{tr}\big((DL)^3\big) &= \sum_{i,j,k} d_i d_j d_k p_i p_j p_k\, e^{-\frac{1}{2}|q_i-q_j|}\, e^{-\frac{1}{2}|q_j-q_k|}\, e^{-\frac{1}{2}|q_k-q_i|}\,.
\end{aligned}
\tag{65}
$$

One gets a "deformed" version of the Camassa–Holm Hamiltonians, with deformation parameters $d_i$. Since they can be interpreted as a rescaling $p_j \to d_j p_j$ of the momenta, the corresponding PDE will have just the same rescaled form.

$T$-**boundary term.** Choosing now $\gamma = DTD$ as a solution to the reflection equation, we get:

$$
\begin{aligned}
\operatorname{tr}(\gamma L) &= \sum_i d_i^2 p_i + 2\sum_{i<j} d_i d_j\, \sqrt{p_i p_j}\, e^{-\frac{1}{2}|q_i-q_j|}\,, \\
\operatorname{tr}\big((\gamma L)^2\big) &= \sum_i d_i^4 p_i^2 + 3\sum_{i\neq j} d_i^2 d_j^2 p_i p_j\, e^{-|q_i-q_j|} + 4\sum_{i\neq j} d_i^3 d_j\, \sqrt{p_i^3 p_j}\, e^{-\frac{1}{2}|q_i-q_j|} \\
&\quad + 6\sum_{\substack{i,j,k \\ \text{all} \neq}} d_i^2 d_j d_k p_i\, \sqrt{p_j p_k}\, e^{-\frac{1}{2}|q_i-q_j|}\, e^{-\frac{1}{2}|q_i-q_k|} \\
&\quad + \sum_{\substack{i,j,k,l \\ \text{all} \neq}} (1+\mathfrak{s}_{jk}\mathfrak{s}_{li})\, d_i d_j d_k d_l\, \sqrt{p_i p_j p_k p_l}\, e^{-\frac{1}{2}|q_i-q_j|}\, e^{-\frac{1}{2}|q_k-q_l|}\,.
\end{aligned}
\tag{66}
$$

Note that since the alternative presentation of Camassa–Holm peakons describes the same Poisson structure, the above Hamiltonians are also valid when using the presentation of section 2.

**$S$-boundary term for $n$ even.** Since for Camassa–Holm peakons, the Lax matrix $L$ is symmetric while $S^\sigma$ is antisymmetric, we get in any Weyl chamber

$$\text{tr}\left((S^\sigma L)^{2m+1}\right) = 0, \forall m. \tag{67}$$

As an example of non-vanishing Hamiltonian, we have for $\gamma = D S^\sigma D$ (in the Weyl chamber defined by $\sigma$):

$$\text{tr}\left((\gamma L)^2\right) = 2 \sum_{\ell=0}^{n/2-1} \bar{d}_{\ell^\circ}^\sigma p_{\sigma(\ell^\circ)} p_{\sigma(\ell^\circ+1)} \left(e^{-|q_{\sigma(\ell^\circ+1)}-q_{\sigma(\ell^\circ)}|} - 1\right), \tag{68}$$

where $\ell^\circ = 2\ell + 1$ and $\bar{d}_{\ell^\circ}^\sigma = d_{\sigma(\ell^\circ)}^2 d_{\sigma(\ell^\circ+1)}^2$.

## 6.2 Degasperis–Procesi Hamiltonians

**Diagonal boundary term.** Considering immediately the case with diagonal matrix $\gamma'$, we have

$$\text{tr}(\gamma' L) = \sum_i d_i p_i,$$

$$\text{tr}\left((\gamma' L)^2\right) = \sum_i d_i^2 p_i^2 + \sum_{i<j} d_i d_j p_i p_j \left(2 - e^{-|q_i-q_j|}\right) e^{-|q_i-q_j|},$$

$$\text{tr}\left((\gamma' L)^3\right) = \sum_{i,j,k} d_i d_j d_k p_i p_j p_k \left(-3 e^{-|q_i-q_k|-|q_j-q_k|} + 4 e^{-\frac{1}{2}(|q_i-q_k|+|q_j-q_k|+|q_j-q_i|)}\right). \tag{69}$$

The "usual" Hamiltonians $\text{tr}\, L^m$ are recovered by setting $d_i = 1, \forall i$.

**$T$-boundary term.** In any Weyl chamber, we get

$$\text{tr}(\gamma' T \gamma' L) = \sum_i d_i p_i + \sum_{i\neq j} d_i d_j \sqrt{p_i p_j} \, e^{-|q_i-q_j|},$$

$$\begin{aligned}
\text{tr}\left((\gamma' T \gamma' L)^2\right) = & \left(\sum_{i,j} d_i d_j \sqrt{p_i p_j}\right)^2 + \sum_{i\neq j} d_i^2 d_j^2 p_i p_j \left(1 - e^{-|q_i-q_j|}\right)^2 \\
& - 4 \sum_{i\neq j} d_i^2 d_j p_i \sqrt{p_j}\left(1 - e^{-|q_i-q_j|}\right)\left(\sum_k d_k \sqrt{p_k}\right) \\
& - 8 \sum_{q_j<q_i<q_k} d_i d_j d_k \sqrt{p_i p_j p_k}\left(1 - e^{-|q_j-q_k|}\right)\left(\sum_l d_l \sqrt{p_l}\right) \\
& + 2 \sum_{\substack{i,j,k \\ \text{all} \neq}} d_i^2 d_j d_k p_i \sqrt{p_j p_k}\left(1 - e^{-|q_i-q_j|}\right)\left(1 - e^{-|q_i-q_k|}\right) \\
& + 8 \sum_{q_i<q_j<q_k<q_l} d_i d_j d_k d_l \sqrt{p_i p_j p_k p_l}\left(1 - e^{-|q_i-q_l|}\right)\left(1 - e^{-|q_j-q_k|}\right).
\end{aligned} \tag{70}$$

**$S$-boundary term for $n$ even.** In the Weyl chamber characterized by $\sigma$, we get for $\gamma'^\sigma = D S^\sigma D$:

$$
\begin{aligned}
\mathrm{tr}(\gamma'^\sigma L) &= 2 \sum_{\ell=0}^{\frac{n}{2}-1} \bar{d}_{\ell^\circ}^\sigma \sqrt{p_{\sigma(\ell^\circ)} p_{\sigma(\ell^\circ+1)}} \left(1 - e^{-|q_{\sigma(\ell^\circ)} - q_{\sigma(\ell^\circ+1)}|}\right), \\
\mathrm{tr}\left((\gamma'^\sigma L)^2\right) &= 2 \sum_{\ell=0}^{\frac{n}{2}-1} (\bar{d}_{\ell^\circ}^\sigma)^2 \, p_{\sigma(\ell^\circ)} p_{\sigma(\ell^\circ+1)} \left(1 - e^{-|q_{\sigma(\ell^\circ)} - q_{\sigma(\ell^\circ+1)}|}\right)^2 \\
&\quad + 2 \sum_{\ell=0}^{\frac{n}{2}-1} \sum_{\substack{j=0 \\ j \neq \ell}}^{\frac{n}{2}-1} \bar{d}_{\ell^\circ}^\sigma \bar{d}_{j^\circ}^\sigma \sqrt{p_{\sigma(\ell^\circ)} p_{\sigma(\ell^\circ+1)} p_{\sigma(j^\circ)} p_{\sigma(j^\circ+1)}} \\
&\quad \times \left(2 e^{-|q_{\sigma(\ell^\circ+1)} - q_{\sigma(j^\circ)}|} - e^{-|q_{\sigma(\ell^\circ+1)} - q_{\sigma(j^\circ+1)}|} - e^{-|q_{\sigma(\ell^\circ)} - q_{\sigma(j^\circ)}|}\right),
\end{aligned}
\tag{71}
$$

where we noted $\ell^\circ = 2\ell + 1$ and $j^\circ = 2j + 1$ to have more compact expressions. Once again, we noted $\bar{d}_{\ell^\circ}^\sigma = d_{\sigma(\ell^\circ)}^2 d_{\sigma(\ell^\circ+1)}^2$.

### 6.3 Novikov Hamiltonians

We recall that the Novikov and Camassa–Holm Lax matrices are related by $L^{Nov} = T L^{CH}$ and that the solutions of the dual reflection equation (60) are related dually: $\gamma'' = \gamma T^{-1}$. The basic combination $\gamma'' L$ entering the Hamiltonians for the Novikov peakons therefore yields the same object as for the Camassa–Holm case. Hence Camassa–Holm and Novikov peakons share identical forms of commuting Hamiltonians, a consistent consequence of their sharing the same Poisson structure (47) after renormalization (46).

## 7 Conclusion

We should start this section with some brief comments on the newly constructed integrable Hamiltonians in section 6, focusing for simplicity on the Camassa–Holm case (66) and (68). They exhibit original features when compared to the already known ones (64): the already commented-on explicit $\sqrt{p}$ dependence; the three- and four-body interaction already present at order 2 in (66) and the pairing of nearest neighbours $2\ell, 2\ell + 1$ observed in (68). It is not clear at this stage whether such features may arise from the description of peakon dynamics when non trivial boundary conditions and/or defects conditions (shock or jump at some fixed point) are introduced in the Camassa–Holm equation. Indeed this type of boundary condition with "reflection" effects usually yields dynamics for the collective modes characterized by potentials depending on $x + y$ which are not directly observed here inside the absolute-value terms. However, the interaction of 3 and 4 peakons at order 2 or the pairing of nearest neighbour peakons may still be a more subtle effect of boundary/defect conditions "entangling" direct and reflected/transmitted peakons: as an example, the 4-body absolute-value terms in (66) always contain two + signs and two − signs, possibly signaling an interaction between two direct and two reflected peakons. However, the nearest neighbour pairing observed in (68) is not obvious to characterize as such. In any case a much more detailed analysis would be required.

Having established the quadratic Poisson structures for three integrable peakon models, in every case based on the Toda molecule $r$-matrix and its partial transposition, many issues remain open or have arisen in the course of our approach.

Amongst them we should mention the problem of finding compatible linear Poisson structures and their underlying $r$-matrix structures. Only solved for the Camassa–Holm peakons

this question is particularly subtle in the case of Degasperis–Procesi peakons as follows from the discussion in Section 3.3.

Another question left for further studies here is the extension of integrability properties beyond the exact peakon dynamics, on the lines of the work in [11] regarding Camassa–Holm peakons and (not surprisingly) based on the linear (and canonical) Poisson structure, yet unidentified in other cases. The difficulty is to disentangle , whenever only a quadratic structure is available, the peakon potential in the Lax matrix from the dynamical weight function in the *n*-body Poisson brackets (the so called *G* function in [10]).

Excitingly, the quadratic structures we have found open the path to a quantized version of peakon models, in the form of *ABCD* algebras, following the lines developed in [5]. We hope to come back on this point in a future work.

Finally in this same extended integrable peakons the still open problem of full understanding of the unavoidably dynamical *r*-matrix structure remains a challenge.

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
