# Peer review of "Integrable quadratic structures in peakon models"

_SciPost Physics, doi:SciPost Phys. 13, 044 (2022)_

## Round 1 · Referee Report · Laszlo Feher (Referee 1) · 2022-4-24

Strengths

  1. The authors found the quadratic Poisson structure obeyed by the peakon Lax matrix $L(q,p)$ as a consequence of the non-canonical Poisson brackets, and this connects the theory of peakons to the general algebraic theory of integrability of finite dimensional Hamiltonian systems.

  2. Several new problems arise from this work: for example the question of quantization by algebraic means, or the question of the existence of linear r-matrix brackets.

  3. The paper is well written, sufficient details of the derivations are presented.

Weaknesses

There are a few typos that should be corrected, and the domain of the $p_i$ variables should be made explicit.

Report

Peakons represent an interesting class of solutions of certain (1+1) dimensional integrable fluid PDEs. The n-peakon solutions are parametrized by 2n variables $q_i, p_i$ that are subject to finite dimensional induced dynamics derived from the field equations. The authors investigate the Poisson structures of the induced dynamics in 3 cases, which correspond to the Camassa-Holm, Degasperis-Procesi and Novikov shallow-water equations. In each case it is already known that the peakon dynamics can be written in Hamiltonian form with the Hamiltonian given by the sum of the $p_i$, and the pertinent Poisson structure has a non-canonical, `non-local’ form. A Lax representation of the peakon dynamics is also known in all 3 cases.

The main content of the paper is the derivation of the quadratic Poisson structure obeyed by the Lax matrix $L(q,p)$ as a consequence of the non-canonical Poisson brackets of the $q_i, p_i$ variables. The $a,b,c,d$ structure elements of the quadratic bracket (1.4) turn out to be non-dynamical, in the sense that they are constant in each open Weyl chamber of the $q$-variables. These are new results, and are important since they lead to an algebraic proof of the Liouville integrability of the peakon dynamics, and may also pave the way for its quantization.

As the authors describe, in the Camassa-Holm case a linear r-matrix structure compatible with the quadratic one is already known, and in the other two cases they find that such a linear structure, having the usual form motivated by the quadratic algebra, does not exist. An additional result of the paper is the derivation of classes of solutions of the dual classical reflection equation (5.1), and the explicit formulae of some ``deformed’’ commuting Hamiltonians based on those solutions.

All in all, I find that the paper is well written and contains interesting results. In particular, it provides a new link between peakon solutions of integrable PDES and the algebraic theory of integrability of finite dimensional Hamiltonian systems. I recommend its publication in Sci. Post. Physics.

Requested changes

I noticed a small number of typing errors:
1. In the formula (1.1) a minus sign seems to be missing in the exponential.
2. In the second line above (1.3) the superscript of $h^{(k)}$ is mistyped.
3. In the third line below (2.13) $s$ does not appear to be defined before; it is probably just $b$ in the current notation.
4. In the second line above (2.17), I do not understand the terminology `associativity’ of the linear Poisson bracket. I suppose it is just the Jacobi identity.
5. The name of Oevel is misspelled in reference [24].

I also have some questions and remarks, and ask the authors to comment on these in the paper:
1. Is it true that the formulae (2.11) and (2.14) define a pair of compatible Poisson brackets on the associative algebra of arbitrary real symmetric matrices?
2. In relation to the square roots in the Lax matrices, can the signs of the $p_i$ change in the peakon dynamics? If yes, does this lead to any subtlety regarding the Lax representation? For example, are the Hamiltonians in (6.3) always real?
3. One should insert the word `commuting’ in front of `Hamiltonians’ in the first line of Section 5.
4. It would be nice to say something about the interpretation of the Hamiltonians of the form $tr(\gamma L)$ that appear in equations (6.2) and (6.3). Could the corresponding deformations of the peakon dynamics also arise from the underlying integrable PDE?

---

## Round 1 · Referee Report · Anonymous (Referee 2) · 2022-5-4

Strengths

  1. The results of the paper show that the peakon Hamiltonians are in involution with respect to the non-canonical Poisson bracket on peakon variables.

  2. The paper also puts the study of peakons within the general formalism of finite-dimensional integrable systems, using its standard algebraic methods and structures (namely here integrable quadratic brackets).

  3. This opens interesting perspectives, for instance concerning the quantisation of the peakon integrable systems.

  4. The paper is overall clear and well-written.

Weaknesses

  1. In my opinion, the notations used to distinguish the three considered cases could be improved a bit in some parts of the text.

  2. It could be helpful to have a discussion of how the quantities sgn$(q_i-q_j)$ are treated and whether Dirac distributions can arise in the computations from their derivatives (see question Q1 below).

Report

This paper studies peakon solutions of integrable 1+1 dimensional PDEs. These solutions follow a particular prescribed dependence on the space variable $x$ and are parametrised by $2n$ functions $(q_i(t),p_i(t))$ of the time variable $t$: the initial PDE then reduces to a system of coupled ODEs obeyed by these functions. In particular, one can then study the integrability properties of this finite-dimensional dynamical system: for instance, it was already established in the literature, for three different choices of the initial PDE (Camassa-Holm, Degasperis-Procesi and Novikov), that the dynamics admits a Lax matrix representation. In this context, the main subject of the paper is to study the Hamiltonian properties of this system and in particular the Poisson bracket of the Lax matrix for these three cases.
The authors succeed in this task by showing that this bracket takes the form of a quadratic integrable structure, which in particular ensures the involution of the conserved quantities extracted from the Lax matrix. Importantly, this result is obtained using a bracket on $(q_i,p_i)$ which is not the canonical one, but rather a more complicated bracket, induced by a choice of Poisson structure in the initial PDE and studied earlier in the literature. Moreover, the matrices $(a,b,c,d)$ characterising the integrable quadratic structure are explicitly spelled-out for each of the three cases considered and are shown to be intimately related to the $r$-matrix of the Toda molecule. The algebraic, Yang-Baxter-like, conditions on these matrices ensuring the Jacobi identity of the bracket are also thoroughly discussed. This quadratic integrable structure forms the main result of the paper and is a new and interesting result.
In the case where the initial PDE is the Camassa-Holm equation, it was already known from previous works that the Lax matrix of the peakon system also satisfies a linear Poisson structure with respect to the canonical bracket on $(q_i,p_i)$. In this case, the canonical bracket is compatible with the non-canonical one mentioned above and the theory belongs to the class of bi-Hamiltonian systems. In this paper, the existence of a linear Poisson bracket of the Lax matrix is investigated for the two other peakon systems arising from the Degasperis-Procesi and Novikov PDEs. It is found that such a linear structure does not exist except for the 2-peakon solution of the Novikov equation, at least for a choice of $r$-matrix inspired by the matrices $(a,b,c,d)$ appearing in the quadratic structure.
Finally, the authors also describe solutions of the reflection equation associated with the three quadratic integrable structures under investigation, allowing them to construct general families of deformed Hamiltonians in involution (which reduce to the standard ones of the peakon system when one considers the trivial unit solution of the reflection equation).

The paper is in general well written and clear. In my opinion, the results obtained are interesting and worthy of a publication, in particular as they put the study of these peakon solutions within the general formalism of finite-dimensional integrable systems, using its standard algebraic methods and structures. In particular, this opens interesting perspectives, including about the quantisation of these systems (e.g. in the spirit of the reference [5] of the paper). I have a few (rather minor) questions on the content of the paper, which are listed below, as well as some comments on notations and typos (see section ''Requested changes''). With the minor revisions these points might require, I recommend this paper for publication in SciPost Physics.

Questions:

Q1. My first question concerns the domain of definition of the coordinates $q_i$ and the type of functional and/or distributional spaces the authors consider. Indeed, the quantities sgn$(q_i-q_j)$ appear in many relevant objects in the paper. It seems to me that, seen as distributions, the derivative of these quantities with respect to a coordinate $q_i$ is naturally proportional to the Dirac distribution $\delta(q_i-q_j)$. Can such ''contact terms'' appear in Poisson brackets? For instance, can it make the brackets with the matrices $(a,b,c,d)$ generally non-vanishing as distributions? And if yes, can this have an effect on the checks for the Jacobi identity? Same question for Poisson brackets with the matrix $T$ in Proposition 4.1. The related point about the domain of the coordinates $q_i$ is whether the authors consider the system as defined on a unique open chamber with fixed signs of $q_i-q_j$ and thus never deal with distributions such as $\delta(q_i-q_j)$? I would like the authors to clarify these points. Moreover, it might be useful for the reader's comprehension and curiosity to mention some of these answers in the article if relevant.

Q2. My second question concerns the discussion around (2.13) and the statement that the Lax matrix considered there is a realisation of the Toda molecule Lax matrix. It naively seems to me that the latter obeys a linear Poisson structure with a $r$-matrix which is completely non-dynamical and thus cannot contain terms like sgn$(q_i-q_j)$. Is this statement true for a specific choice of chamber for the $q_i$'s? If yes, it could be helpful to specify this in the paper.

Q3. This third question is quite minor and the authors should feel free to take it into account or not as it does not affect at all the results and readability of the paper. It is said that the bracket (4.15) can be consistently completed by taking $\lbrace \bar{q}_1+\bar{q}_2,X \rbrace=0$ for all $X$. This makes the Poisson structure degenerate. Is the physical integrable system then ultimately defined on 2-dimensional symplectic leaves? Is there another consistent choice of bracket for $\bar{q}_1+\bar{q}_2$ making the structure symplectic?

Requested changes

I have listed some questions on the content of the paper in the section ''Report''. Depending on the answers of the authors, these can suggest some minor revisions and adding explanations in the paper.

The next two comments are about notations and conventions.

  1. My main remark on notations concerns the way the three different cases treated in the paper are distinguished, as I found it slightly confusing in some places. The same notation $L$ is generally used for the three Lax matrices and a label $CH$ or $N$ is sometimes added when there is need to distinguish the cases. On the other hand, the authors write $a_{12}$, $a'_{12}$ and $a''_{12}$ for the matrices in the Poisson structure in the three cases (with $a_{12}$ also sometimes referring to an unspecified matrix in the general discussion of quadratic brackets). In my opinion, it could be useful to be a bit more precise and uniform in the notations at least in some places, although I understand that the goal was to avoid having heavy labelling of many objects. In particular, in the 2nd paragraph of section 5: is the solution $\gamma$ there associated with the Degasperis-Procesi matrices $(a',b',c',d')$ while the solution $\gamma'$ is associated with the Novikov matrices $(a'',b'',c'',d'')$? If yes, it would seem relevant to change $\gamma$ and $\gamma'$ to $\gamma'$ and $\gamma''$ respectively. In this same paragraph, should $c_{12}=b_{21}'$ and $c_{12}=b_{21}''$ in fact read $c_{12}'=b_{21}'$ and $c_{12}''=b_{21}''$? Overall, my impression is that slightly heavier but more precise notations, for instance with labels $CH$ and $N$, can help the reader easily identify the different relevant objects and also distinguish the explicit examples with the general discussions (it seems to me however that this is a rather subjective point and thus should in the end be left to the judgement of the authors).

  2. A second remark around equation (2.13): it could be useful to explain quickly the notations and conventions for $h_i,e_\alpha,x_i,x_\alpha,\Delta_+,\alpha(i), \dots$ In particular, are the Cartan elements $h_i$ chosen here to be orthonormal with respect to the bilinear trace form and are the root vectors $e_\alpha$ normalised to have a 1 in their non-trivial entry as matrices? It also seems to me that in the second Poisson bracket after (2.13), $h_i$ should be $x_i$.

The rest of the comments concern typos and very minor corrections.

  1. Before equation (1.3), $h^{k)}$ should be $h^{(k)}$.
  2. $\mathfrak{s}_{ij}$ is defined after equation (2.8) but is already used in (2.3).
  3. It could be useful to explicitly define the notation $t_2$ when it first appears in (2.12).
  4. $e_{ij}$ is defined after equation (2.12) but is already used in (2.10).
  5. In the last paragraph of section 2.1, $s$ should be $b$.
  6. In Remark 3.1, it is said that the pair of matrices $(a',b')$ yields an alternative representation for the Camassa-Holm peakons with matrices $(a,b+\frac{1}{4}\mathcal{Q})$. Is this up to a factor 2?
  7. In (3.16), should there be either a different sign for the second matrix $a'_{12}$ or an exchange of indices $a'_{21}$?
  8. In the second paragraph of section 5, it is said that $a'{12}=d'$}=4a_{12. Should the factor 4 be in fact a 2 (see equation (3.9))?

---

## Round 2 · Referee Report · Anonymous · 2022-6-14

Report

This is the second version of the paper, which deals with integrable Poisson structures in peakon dynamics. The authors have answered the questions and implemented the suggestions and corrections raised in my report on the first version. As mentioned in the first report, the paper successfully relates the study of peakons with the standard algebraic methods of integrability, such as quadratic integrable Poisson structures, and opens interesting perspectives, for instance on the quantisation of these models. Taking that into account, and with the corrections implemented in this version 2, I am happy to recommend this article for publication in SciPost Physics.

---

## Round 2 · Referee Report · Laszlo Feher · 2022-6-16

Report

In this revised version the authors have answered the questions and comments that I made in my report on the original manuscript. By this and other minor corrections and additions they have improved the quality of the paper. Thus I have no hesitation to recommend its publication in SciPost Physics.

---

## Round 2 · List of Changes

Dear Editor,

We have revised our manuscript according to the requested changes of the referees. Here are the major modifications:

- page 3 [section Introduction]
a/ we specified the domain of validity of our study regarding the problem of the boundaries of the Weyl chambers raised by question Q1 of referee 2.
b/ as suggested by the second referee, we have fixed the notation for the matrices appearing in the different models (introducing prime and double prime notation), see comment 1 of referee 2.

- page 5 [section 2.1 Linear Poisson structure]
a/ we made clear the validity of the identification with the r-matrix of Toda molecule after ref. 22, see question Q2 of referee 2.
b/ we explained quickly the notations and conventions for the Lie algebra elements h_i, e_\alpha, etc. after eq. (2.13), see comment 2 of referee 2.

- page 6
The first paragraph of page 6 has been added following the remark 1 of referee 1 about compatible PB on the associative algebra of arbitrary real symmetric matrices ("The Poisson bracket structure (2.11) is indeed ...).

- page 12
Last paragraph before section 4.3 : we commented on the degeneracy of the PB structure obtained in (4.15), see comment 3 of referee 2.

- page 13 [section 5]
In order to comment on remark 4 of referee 1, we expanded the beginning of section 5 (until formula 5.4 included), by pointing out the existence of two objects in the general construction of Poisson commuting Hamiltonians from quadratic structures.
A sentence has also been added after eq. (6.2) on page 15.

Moreover, remark 5.1 on page 14 has been rephrased to clarify the point.

- page 15 [section 6]
we answer the comment 2 of referee 1 on the sign of p_i in relation to the square roots in the Lax matrix.

- page 18 [conclusion]
The conclusion has been expanded with a new paragraph (the first one) by commenting on the newly constructed integrable Hamiltonians, see e.g. comment 4 of referee 1.

Finally, we corrected the typos and done the minors corrections raised by the referees.

Best regards,
The authors

---

## Editorial Decision

published